Early Agenian rhinocerotids from Wischberg (Canton Bern, Switzerland) and clarification of the systematics of the genus Diaceratherium

Jame Claire 1
http://orcid.org/0000-0002-8517-1612 Tissier Jérémy 2 3 jeremy.tissier@unifr.ch
http://orcid.org/0000-0002-0956-0712 Maridet Olivier 2 3
Becker Damien 2 3
1 Observatoire des Sciences de l’Univers de Rennes, Université Rennes I , Rennes , France
2 Cenozoic Research Group, JURASSICA Museum , Porrentruy , Switzerland
3 Department of Geosciences, University of Fribourg , Fribourg , Switzerland
Stefen Clara
Electronic publication date: 2019 Aug 28
Publication date: 2019
Volume: 7
Electronic Location ID: e7517
Received 2019 May 16; Accepted 2019 Jul 19
Copyright: © 2019 Jame et al.
Copyright year: 2019
Copyright holder: Jame et al.
License: This is an open access article distributed under the terms of the Creative Commons Attribution License, which permits unrestricted use, distribution, reproduction and adaptation in any medium and for any purpose provided that it is properly attributed. For attribution, the original author(s), title, publication source (PeerJ) and either DOI or URL of the article must be cited.
License URL: https://creativecommons.org/licenses/by/4.0/

Keywords: Paleontology, Rhinocerotidae, Agenian, Switzerland, Diaceratherium, Pleuroceros, Systematics, Ecology, Anatomy, Miocene

Funding: Swiss National Science Foundation 200021_162359 This project was financially supported by the Swiss National Science Foundation (projects 200021_162359). The funders had no role in study design, data collection and analysis, decision to publish, or preparation of the manuscript.

==============================
Background

Wischberg is a Swiss locality in Bern Canton which has yielded numerous vertebrates remains from the earliest Miocene (= MN1). It has a very rich faunal diversity, one of the richest in Switzerland for this age. Among all the mammals reported in the original faunal list 70 years ago, three rhinocerotid species were identified. The material consists of two fragmentary skulls, cranial fragments, several mandibles, teeth and postcranial bones, in a rather good state of preservation.

Results

After reexamination of the material from this locality (curated in three different Swiss museums) and comparison with holotype specimens, we show that all rhinocerotid specimens from Wischberg can be referred to two species only. Most of the material can be attributed to the large-sized teleoceratine Diaceratherium lemanense, while only a few specimens, including a skull and mandible, belong to the much smaller sized Pleuroceros pleuroceros. We describe and illustrate for the first time most of these fossil remains. However, the systematics of the genus Diaceratherium is currently controversial, and based on our new observations we consider seven species as valid, though a large-scale phylogenetic study should be done in the future to resolve it. The rhinocerotid association found in Wischberg is nonetheless typical of the MN1 biozone, which results from a faunal renewal occurring just before the end of the Oligocene.

Introduction

The Aquitanian Lower Freshwater Molasse (“Untere Süsswassermolasse”) record of the Plateau Molasse is characterized within the central and eastern area of the Swiss north alpine foreland basin by the floodplain deposits from the Granitische Molasse formation, lateral equivalent of the Molasse grise de Lausanne formation from the western area (Habicht, 1987; Berger et al., 2005a, 2005b; Schweizerisches Komitee für Stratigraphie und Landesgeologie, 2014). These geological formations yielded many vertebrate localities, unfortunately recording mostly incomplete assemblages and only a few large mammal species (Scherler et al., 2013). However, Agenian land mammal associations are remarkably well documented in the localities of Wischberg (Aquitanian age, MN1 biozone; Schaub & Hürzeler, 1948; Engesser & Mödden, 1997), Engehalde (MN2; Becker et al., 2010) and Wallenried (MN2; Becker et al., 2001; Mennecart et al., 2016).

From the area of Langenthal (Bern Canton, Switzerland), Gerber (1932, 1936) first reported fossil rhinocerotids originating from the Wischberg locality (latitude 47.199157894°/longitude 7.763943664°; Fig. 1). A preliminary mammal list was provided by Schaub & Hürzeler (1948), including Eulipotyphla, Rodentia, Lagomorpha, Cainotheriidae, non-ruminant Artiodactyla, Ruminantia, Tapiridae and Rhinocerotidae. More recently, Lagomorpha have been reviewed by Tobien (1975) and part of large mammals by Becker (2003) and Scherler, Becker & Berger (2011) and Scherler et al. (2013). Since the work of Engesser & Mödden (1997) on the mammal biozonation of the Lower Freshwater Molasse of Switzerland, the mammal assemblage of Wischberg (Table 1) can be considered as one of the most important and complete in the Swiss Molasse Basin, consistently pointing to an early Aquitanian age.

Figure 1 General setting of Wischberg locality, Bern Canton, Swiss Molasse basin (MN1, Agenian, earliest Miocene).

(A) Map of a part of Western Europe showing the location of Switzerland. (B) Enlargement of the Aquitanian paleogeographical context of the Swiss Molasse Basin, with detailed location of Wischberg locality (star symbol).

Table 1 Mammal assemblage of Wischberg locality, Bern Canton, Swiss Molasse basin (MN1, Agenian, earliest Miocene).

After Schaub & Hürzeler (1948)	After Tobien (1975), Scherler et al. (2013) and this study	
Talpidarum indet.	Talpidae indet.	
Erinaceus priscus	Amphechinus edwardsi	
Lagomorphum aff. piezodus	Piezodus tomerdingensis	
Cricetodon cf. hochheimensis	Eucricetodon cf. hochheimensis	
Cricetodon collatus	Eucricetodon collatus	
Plesiosminthus myarion	Plesiosminthus myarion	
Rhodanomys schlosseri	Rhodanomys schlosseri	
Rhodanomys sp. nov.	Rhodanomys sp. nov.	
Eomyidarum gen. nov.	Ritteneria sp.	
Gliridarum gen. nov.	Gliridae indet.	
Cainotherium laticurvatum	Cainotherium laticurvatum	
Elomeryx minor	Elomeryx minor	
Palaeochoerus meissneri	Hyotherium meissneri	
Amphitragulus sp.	Amphitragulus elegans	
Tapirus intermedius var. robustus	Eotapirus broennimanni (adult specimens)	
Tapirus brönnimanni	Eotapirus broennimanni (juvenile specimens)	
Aceratherium lemanense	Diaceratherium lemanense	
Diceratherium asphaltense	Diaceratherium lemanense	
Diceratherium pleuroceros	Pleuroceros pleuroceros	

In this work, we first review the description and the identifications of the rhinocerotid material from Wischberg, which were assigned to three species by Schaub & Hürzeler (1948): the single-horned and short-limbed teleoceratines Diaceratherium lemanense (Pomel, 1853) and D. asphaltense (Depéret & Douxami, 1902), as well as the small-sized tandem-horned Pleuroceros pleuroceros (Duvernoy, 1853). Second, we examine the systematics of the genus Diaceratherium, which is currently contentious, and the ecological role of the Early Miocene Rhinocerotidae within the large herbivorous mammal communities of Western Europe.

Materials and Methods

The fossil materials from Wischberg were discovered between 1931 and 1947 in two pits of Aquitanian mottled marls and sands of the Granitische Molasse (Schaub & Hürzeler, 1948) that were exploited during the first half of the last century in Langenthal (Bern Canton, Switzerland). The sites are no longer accessible due to anthropogenic developments. The studied material includes 24 rhinocerotid specimens (and among them three casts) that are stored in the natural history museums of Bern (Naturhistorisches Museum des Burgergemeinde Bern) and Basel (Naturhistorisches Museum Basel), as well as in the local museum of Langenthal (where the original skull and a mandible of D. lemanense are exposed). It is worth to clarify that the original specimens referred to Pleuroceros pleuroceros, except the semilunate NMBE5031537, are lost and the work on this taxon is based on the remaining casts.

The rhinocerotid specimens from Wischberg have been described by means of anatomical descriptions, comparative anatomy and biometrical measurements. The sequence of described dental and osteological features follows Antoine (2002). The dental terminology follows Heissig (1969) and Antoine (2002), while dental and skeletal measurements were taken according to Guérin (1980). The locomotion type is based on the gracility index of the McIII and MtIII (100 × TDdia/L; Guérin, 1980).

The stratigraphical framework is based on geological time scales and European Land Mammal Ages for the Neogene (Hilgen, Lourense & Van Dam, 2012). Successions of Mammal Neogene units were correlated by Berger (2011) based on biostratigraphic and magnetostratigraphic data (BiochroM’97, 1997; Engesser & Mödden, 1997; Kempf et al., 1997, 1999; Mein, 1999; Steininger, 1999; Agustí et al., 2001).

Body masses of the rhinocerotid species found in Wischberg are estimated from dental and postcranial measurements. The equations used to estimate the body mass of rhinocerotids are based on the correlations established for perissodactyls by Legendre (1989), for Rhinocerotidae by Fortelius & Kappelman (1993) and for mammals by Tsubamoto (2014).

Results

Systematic paleontology

Class Mammalia Linnaeus, 1758

Order Perissodactyla Owen, 1848

Superfamily Rhinocerotoidea Gray, 1821

Family Rhinocerotidae Gray, 1821

Subfamily Rhinocerotinae Gray, 1821

Genus Pleuroceros Roger, 1898

Type species: Pleuroceros pleuroceros (Duvernoy, 1853)

Included species: Pleuroceros blandfordi (Lydekker, 1884)

Pleuroceros pleuroceros (Duvernoy, 1853)

Figs. 2 and 3; Tables 2–4

Stratigraphical range: Latest Oligocene (?MP29/30) to Early Miocene (MN1–MN2), western and central Europe (Antoine & Becker, 2013).

Occurrences: - France: Billy-Base (Allier), ?MN29/30; Gannat, MN1 (type locality); Paulhiac, MN1; Pyrimont-Challonges, MN1; Saulcet, MN1; Laugnac, MN2; Montaigu-le-Blin, MN2; (Duvernoy, 1853; Lavocat, 1951; De Bonis, 1973; Hugueney, 1997; Ginsburg & Bulot, 2000; Antoine et al., 2010; Antoine & Becker, 2013; Scherler et al., 2013).

- Germany: Flörsheim, MN2; Pappenheim, MN2 (Schlosser, 1902; Heissig, 1999).

- Switzerland: Wischberg, MN1 (Schaub & Hürzeler, 1948; Heissig, 1999; Becker, 2003).

Referred material: Skull with right P1-M3 and left P2-M3 (original specimen lost, cast NMBE5031553, cast NMB-AS77), fragmented mandible with right p4-m3 and left m1–2 (original lost, cast NMBE5026739, cast NMB-AS78), right semilunate (NMBE5031537, cast NMB-AS3), right McIV (original lost, cast NMB-AS79) from Wischberg (Switzerland, MN1).

Description

Skull. NMBE5031553 is a cast of an incomplete, fragmented and transversally compressed skull comprising a part of the frontals, the area of the right zygomatic arch, the right P1-M3 and the left P2-M3. Few cranial characters are observable. We can note a remarkably curved upwards jugal bearing a processus postorbitalis, an infraorbital foramen situated above the P3, an anterior border of the orbit reaching the level of the paracone of M1, an anterior base of the zygomatic process high above the M1, and the presence of a processus lacrimalis (Fig. 2A).

Figure 2 Pleuroceros pleuroceros (Perissodactyla, Rhinocerotidae) from Wischberg locality, Bern Canton, Swiss Molasse basin (MN1, Agenian, earliest Miocene).

Partial skull NMBE5031553 in lateral (A), dorsal (B), medial (C) and occlusal (D) views and left-side fragment with P2-M3 from the same individual in occlusal (E) view. Mandible fragments NMBE5026739 in labial (F, G), lingual (H, I) and occlusal (J, K) views with p4-m3 (right-side fragment) and m1–2 (left-side fragment). Photo credit: Patrick Röschli.

Mandible. From the fragmented mandible NMBE5026739, the corpus mandibulae (height below m3 = 71.5 mm) does not seem to bear a median sagittal groove (sulcus mylohyoideus). The retromolar space is short and the position of the foramen mandibulae (based on the transverse slimming of the corpus in cross section) is located below the alveolar level (Figs. 2F and 2G).

Upper teeth. The dental wear of the tooth series is advanced (Figs. 2D and 2E). The premolars are not reduced compared with the molars (LP3–4/LM1–3 > 50; Table 2). The dental structures are simple, without secondary enamel folds. The cheek teeth are brachydont (low-crowned), and the roots are long and distinct. The upper cheek teeth lack crista and medifossette. The paracone fold is present on all cheek teeth and strong on lesser worn teeth such as the M2–3. The premolars are molariform (sensu Heissig, 1969) and lack any crochet, antecrochet and constriction of both protoloph and metaloph. The labial cingulum is reduced to the posterior part of the ectoloph and the lingual cingulum is reduced to the opening of the median valley. On P2–4 the postfossette is narrow and the metaloph is posterolingually oriented. The P1 is much narrower than P2 and triangular in occlusal view (Fig. 2D). On P2, the protocone is as developed as the hypocone, the metaloph is directed posterolingually, and the protoloph is continuous and widely connected with the ectoloph. A crochet is always present on upper molars, but the metaloph is not constricted. The labial cingulum is weak and absent at the base of the paracone fold, whereas the lingual cingulum is reduced to the base of the posterior half of the protocone, reaching the opening of the median valley. The metastyle is long and the metacone fold is absent. On M1–2, the protoloph is slightly constricted and it bears an antecrochet, the metaloph is short and the distal part of the ectoloph is straight. A weak mesostyle is present on M2. The M3 has a roughly triangular occlusal outline, though the ectoloph and metaloph are fused in a characteristic convex ectometaloph without posterior groove. The protoloph is not lingually elongated, without constriction and antecrochet.

Table 2 Dimensions of the cheek teeth of Pleuroceros pleuroceros (Perissodactyla, Rhinocerotidae) from Wischberg locality, Bern Canton, Swiss Molasse basin (MN1, Agenian, earliest Miocene).

Casts NMBE5031553 and NMB-AS77	Casts NMBE5026739 and NMB-AS78	
Upper tooth row	LP3–4	LM1–3	LP3–4/LM1–3 × 100	Lower tooth row	Lp3–4	Lm1–3	Lp3–4/Lm1–3 × 100	
Left	53.5	94.0	56.9					
Right	54.0	95.0	56.8	Right	–	101.5	–	
Upper cheek teeth	L	W	H	Lower cheek teeth	L	W	H	
Right P1	15.1	15.1	–					
Left P2	23.2	26.8	–					
Right P2	24.0	27.1	–					
Left P3	25.7	34.6	–					
Right P3	27.8	36.6	–					
Left P4	27.8	37.8	–					
Right P4	27.1	37.2	–	Right p4	28.0	19.9		
Left M1	31.8	38.1	–	Left m1	30.5	18.2		
Right M1	31.0	35.8	–	Right m1	29.0	(19.0)		
Left M2	37.5	40.3	20.2	Left m2	34.5	21.3		
Right M2	39.0	41.3	19.1	Right m2	33.6	21.0		
Left M3	32.0	37.5	23.7					
Right M3	33.8	38.3	–	Right m3	36.9	20.8		
Note:

Dimensions are given in mm and those in parentheses are estimated.

Lower teeth. On lower cheek teeth, the labial cingulum is reduced to a thin bulge at the base of the external groove and the lingual one is completely absent (Figs. 2F and 2G). The external groove is developed and is vanishing above the neck. The trigonid is angular and forms a right dihedron in occlusal view (Fig. 2H). The metaconid and the entoconid are not constricted. The posterior valley is V-shaped, but wider on the lower molars than on the premolars. The hypolophid of the lower molars is oblique and there is no lingual groove on the entoconid of m2–3.

Semilunate. The semilunate NMBE5031537 (Figs. 3A–3E) is slightly rolled and eroded (Table 3). The medial and lateral facets are not preserved, except for the flat, ovoid and sagittally elongated proximomedial facet for the scaphoid (Fig. 3E). In proximal view (Fig. 3B), the ulna-facet is lacking and, in anterior view (Fig. 3A), the anterior side is smooth with an acute distal border, high and narrow. The proximal facet is very convex and short sagittally (Fig. 3B). The magnum-facet is roughly flat in its anterior half and concave posteriorly (Fig. 3E).

Table 3 Dimensions of the semilunate of Pleuroceros pleuroceros from Wischberg (NMBE5031537) and comparison with other specimens of Pleuroceros and Diaceratherium.

Semilunate	TD	APD	H	
P. pleuroceros
Wischberg (this study)	30.6	53.2	38.8	
P. pleuroceros
Paulhiac (De Bonis, 1973)	(34.5)	(45.0)	(34.5)	
P. blandfordi
Bugti Hills (Antoine et al., 2010)	34.0	55.0	36.0	
D. lamilloquense
Castelmaurou (Duranthon, 1990)	43.8	52.5	47.5	
D. asphaltense
Pyrimont-Challonges (Type, coll. UCBL)	44.8	59.5	43.1	
D. asphaltense
Saulcet (coll. NMB)	47.0–47.0	67.0–67.3	51.0–51.5	
D. aginense
Laugnac (coll. MHNM)	38.0–44.4
40.7 [8]	60.7–65.0
62.1 [6]	45.6–51.5
48.4 [8]	
D. aurelianense
Artenay (Cerdeño, 1993)	34.5	67.3	67.3	
Note:

Dimensions are given in mm, those in parentheses are estimated and those in italics are based on the literature. Localities are indicated below the taxon name and those in bold font are the type localities of the species. The upper line indicates the minimum and maximum dimensions, and the bottom line (when several specimen are used) indicates the average value and the number of specimens (in brackets).

Figure 3 Pleuroceros pleuroceros (Perissodactyla, Rhinocerotidae) from Wischberg locality, Bern Canton, Swiss Molasse basin (MN1, Agenian, earliest Miocene).

Right semilunate NMBE5031537 in anterior (A), proximal (B), distal (C), lateral (D) and medial (E) views and right McIV (cast NMB-AS79) in anterior (F), lateral (G), posterior (H), medial (I) and proximal with dorsal toward top (J) views. Photo credit: Patrick Röschli.

Metacarpals. The McIV NMB-AS79 (Figs. 3F–3J) is short and rather gracile (GI = 23.0; Table 4), sagittally flattened, with a short insertion for the m. interossei on the medial side (Fig. 3I). It bears a salient insertion of the m. extensor carpalis on the anterior side (Fig. 3F), and a high and acute median keel of the distal articulation. In proximal view, the proximal facet is trapezoidal (Fig. 3J) and the articulation facet for the McV on the lateral side is not preserved (Fig. 3G).

Table 4 Dimensions of the McIV of Pleuroceros pleuroceros from Wischberg (NMB AS79) and comparison with Diaceratherium species (Perissodactyla, Rhinocerotidae).

McIV	L	TD
prox	APD
prox	TD
dia	APD
dia	TD
dist	APD
dist	
P. pleuroceros
Wischberg (this study)	112.3	32.6	31.1	26.0	15.2	–	28.8	
D. lemanense
Gannat (coll. NMB)	132.5	47.0	39.0	33.0	19.5	43.5	40.5	
P. pleuroceros
Paulhiac (De Bonis, 1973)	(112.5)	(28.5)	(31.5)	(27.0)	(16.0)	(36.5)	(27.0)	
D. lamilloquense
La Milloque (Michel, 1983)	116.0	26.0	–	28.0	–	(30.5)	–	
D. lamilloquense
Castelmaurou (Duranthon, 1990)	125.3	37.4–38.0
37.7 [2]	40.2–41.2
40.7 [2]	32.2–33.3
32.3 [2]	19.9–20.5
20.2 [2]	43.6	37.0	
D. tomerdingense
Tomerdingen (Type, coll. SMNS)	116.3	48.9	39.7	37.0	21.5	43.5	39.3	
D. asphaltense
Pyrimont-Challonges (Type, coll. UCBL)	117.0–122.0
119.5 [2]	40.0–40.5
40.3 [2]	38.5–39.0
38.8 [2]	29.5–30.0
29.8 [2]	19.0–19.5
19.3 [2]	39.0–41.0
40.0 [2]	34.5–35.0
34.8 [2]	
D. asphaltense
Saulcet (coll. NMB)	124.0	34.7	39.5	32.2	21.6	38.5	34.2	
D. aginense
Laugnac (coll. MHNM)	112.2–120.4
117.4 [4]	39.5–40.8
40.2 [2]	43.3–43.6
43.5 [2]	27.3–29.6
28.1 [4]	17.3–17.9
17.7 [4]	38.7–40.4
39.5 [3]	36.0–38.0
37.2 [4]	
D. aurelianense
Neuville-aux-Bois (Cerdeño, 1993)	106.0	42.0	44.6	29.5	18.5	43.8	40.3	
Note:

Dimensions are given in mm, those in parentheses are estimated and those in italics are based on the literature. Localities are indicated below the taxon name and those in bold font are the type localities of the species. The upper line indicates the minimum and maximum dimensions, and the bottom line (when several specimen are used) indicates the average value and the number of specimens (in brackets).

Remarks

Based on comparison with coeval rhinocerotid genera, the referred specimens point to a remarkably small rhinoceros, excluding its assignation to the teleoceratine Diaceratherium. Moreover, this genus differs by a developed external groove and a rounded trigonid on the whole lower cheek tooth series. The specimens differ from the acerathere (sensu lato) Mesaceratherium Heissig, 1969 by the absence of a lingual bridge between the protocone and the hypocone of the upper premolars, the absence of continuous lingual cingulum on P2–4, by a straight posterior part of the ectoloph on M1–2, as well as an oblique hypolophid on lower cheek teeth and a trapezoidal outline of the proximal facet of the McIV (Heissig, 1969; De Bonis, 1973; Antoine et al., 2010). The material from Wischberg differs from Protaceratherium minutum (Cuvier, 1822) by a less angular and V-shaped external groove on lower cheek teeth, as well as the lack of a labial and continuous lingual cingulum, the absence of crochet and crista on upper premolars, and a shorter and stouter McIV (Roman, 1924).

The specimens share with the genus Pleuroceros some morphological similarities, such as a reduced lingual cingulum on upper premolars, the absence of antecrochet on P2–3 and a straight posterior part of the ectoloph on M1–2 (Antoine et al., 2010). The referred specimens differ from Pleuroceros blanfordi (Lydekker, 1884) by ca. 15% smaller size, the absence of a lingual bridge on P2–4 (semimolariform upper premolars, sensu Heissig, 1969), a posteriorly directed metaloph and a hypocone as strong as the protocone on P2, a protocone not constricted on P3–4, the absence of antecrochet on P4, the absence of mesostyle on M2, the metaconid not constricted on lower cheek teeth, and a reduced lingual cingulum on lower premolars (Antoine et al., 2010). The dimensions as well as the postcranial, cranial and dental morphology of Wischberg specimens are in fact extremely similar to the type material and other specimens of Pleuroceros pleuroceros (Duvernoy, 1853) from Gannat (type locality, collection MNHN), notably by the shape of the jugal bearing a processus postorbitalis, the molariform upper premolars lacking antecrochet, the only slightly constricted protoloph on M1–2, the typically convex ectometaloph of M3, the absence of antecrochet and protocone constriction on the M3, the reduction of the labial cingulum, the rather smooth external groove and rounded trigonid on lower cheek teeth and a somewhat short and gracile McIV (Duvernoy, 1853; De Bonis, 1973; Antoine et al., 2010; J. Tissier, 2018, personal observation; Table 4).

Genus Diaceratherium Dietrich, 1931

Original diagnosis (Dietrich, 1931; translation by the authors): “Medium-sized rhinoceros with pneumatised cranial bones; with long, thin and unfused nasal bones; onset of formation of a terminal horn. Four-fingered hand. Mesatipody. Brachyodont. Anterior dentition with large I1/i2. Molars homodont, simple, poorly folded. Decidual dentition: long lasting DI1DI2—DP1–4/di1di2—dp1–4. Permanent dentition: I1—P2–4 M1–3/i2—p2–4 m1–3. Enamel slightly wrinkled, mostly vertically rugged.”

Emended diagnosis: Medium-sized mediportal rhinoceros with long, thin and unfused nasal bones that can bear a small terminal horn. U-shaped nasal notch with a posterior border above P3 and straight occipital crest in dorsal view (not visible on the type material of D. tomerdingense). Anterior dentition with large I1/i2. Decidual dentition: DI1DI2—DP1–4/di1di2—dp1–4. Permanent dentition: I1—P2–4 M1–3/i2—p2–4 m1–3. Upper premolars semi-molariform to molariform with strong lingual cingulum. Upper molars without crista, but with a crochet, antecrochet and reduced lingual cingulum. Enamel slightly wrinkled, mostly vertically rugged. Four-fingered hand.

Type species: D. tomerdingense Dietrich, 1931

Included species: D. lemanense (Pomel, 1853), D. aurelianense (Nouel, 1866), D. asphaltense (Depéret & Douxami, 1902), D. aginense (Répelin, 1917), D. lamilloquense Michel in Brunet, De Bonis & Michel, 1987, D. askazansorense Kordikova, 2001.

Diaceratherium lemanense (Pomel, 1853)

Figs. 4–7; Tables 5–10

Emended diagnosis: Based on comparisons with the material from type localities: Gannat for D. lemanense (MNHN collection and Boada-Saña, Hervet & Antoine, 2007; Boada-Saña, 2008), La Milloque for D. lamilloquense (Michel, 1983), Pyrimont-Challonges for D. asphaltense (UCBL collection), Laugnac for D. aginense (MHNM and UCBL collection), Neuville-aux-Bois for D. aurelianense (MNHN collection and Mayet, 1908; Cerdeño, 1993) and Askazansor for D. askazansorense (Kordikova, 2001).

Table 5 Dimensions of the anterior teeth of Diaceratherium lemanense (Perissodactyla, Rhinocerotidae) from Wischberg locality, Bern Canton, Swiss Molasse basin (MN1, Agenian, earliest Miocene).

Diaceratherium lemanense	
Upper incisors (I1)	APD	TD	H	Lower incisors (i2)	APD	TD	H	
NMBE5031540 (left)	50.2	18.5	18.2	NMBE5031547 (left)	–	–	43.0	
NMBE5031540 (right)	–	17.5	17.1	NMBE5026738 (right)	31.9	24.0	41.2	
NMBE5031546 (right)	–	17	16.0					
Note:

Dimensions are given in mm.

Table 6 Dimensions of the upper cheek teeth of Diaceratherium lemanense (Perissodactyla, Rhinocerotidae) from Wischberg locality, Bern Canton, Swiss Molasse basin (MN1, Agenian, earliest Miocene).

Original NMBE5031539, casts NMBE5031538 and NMB-AS75	
Upper tooth row	LP3–4	LM1–3	LP3–4/LM1–3 × 100	
Right	(68.0)	126.9	(53.6)	
Upper cheek teeth	L	W		
Right P4	(34.5)	(42.6)		
Left M1	39.2	–		
Right M1	39.7	47.0		
Left M2	47.1	51.1		
Right M2	44.0	50.5		
Left M3	48.0	52.6		
Right M3	46.1	–		
Note:

Dimensions are given in mm and those in parentheses are estimated.

Diaceratherium lemanense differs from D. tomerdingense by a larger size, the presence of a posterolingual groove on the protocone of P3–4, a high, elongated and narrow anterior side of the semilunate, the symmetrical proximal border of the trapezoid in anterior view, the presence of a trapezium-facet and a large posterior McIII-facet on the McII, and a band-shaped magnum-facet on the McII. However, the large contact between the McV-facet and the pyramidal-facet of the unciform was believed to be a diagnostic character of D. lemanense, but it is also present in D. tomerdingense.

It differs from D. lamilloquense by the absence of a lingual bridge between the protocone and the hypocone of the upper premolars, a protoloph of P2 more transverse and connected to the ectoloph, the more reduced lingual cingulum on the upper molars and the pentagonal proximal facet of the McIV in proximal view.

It differs from D. asphaltense by the absence of a lingual bridge between the protocone and the hypocone of P2–3, shorter nasals, occipital condyles about 20–25% wider, a magnum with a curved and not straight posterior tuberosity, a high, elongated and narrow anterior side of the semilunate, a band-shaped magnum-facet on the McII, a pentagonal proximal facet of the McIV in proximal view, the presence of an articulation facet for the tibia on the calcaneus and an elongated tuber calcanei in posterior view, and an acute median keels of the distal articulation of the metapodials.

It differs from D. aginense by a square P2 with a protoloph as long as the metaloph, the absence of metacone fold on P3–4, the absence of anterior groove on the protoloph of P3–4, the absence of lingual bridge between the protocone and the hypocone on P2–4, the strong and continuous lingual cingulum on upper premolars, the symmetrical distal articulation of the pyramidal for the semilunate, the greater posterior height of the scaphoid compared to its anterior height, the contact between the McV-facet and the pyramidal-facet of the unciform, a pentagonal proximal facet of the McIV in proximal view and the slender tuber calcanei in posterior view of the calcaneus.

It differs from D. aurelianense by the absence of a postorbital process of the frontals, the more widely separated protocone and hypocone on P2, the stronger and continuous lingual cingulum on P2–4, the absence of anterior groove on the protoloph of P3–4, the absence of lingual cingulum in the openings of the valleys of the lower molars, the contact between the McV-facet and the pyramidal-facet of the unciform, the much longer and more gracile metapodials, the absence of a fibula-facet on the calcaneum and the slender tuber calcanei in posterior view, the higher and narrower astragalus as well as the shorter collum tali and a more concave Cc1 facet.

Finally, it differs from D. askazansorense by a higher and narrower astragal with a much shorter collum tali and a longer and more slender tuber calcanei.

Stratigraphical range: Latest Oligocene (MP30) to Early Miocene (MN2), Western Europe (Antoine & Becker, 2013).

Occurrences: See Table 11.

Referred material: Skull with left M1–M3 (original exposed in ML, cast NMBE5031538, cast NMB-AS75), right maxillary fragment with P3-M3 (NMBE5031539), right and left I1 (NMBE5031540), dental fragments of right I1 (NMBE5031546), left i2 (NMBE5031547), right P1 (NMBE5031548), left P3 (NMBE5031549), right P3 (NMBE5031550), two left lower cheek teeth (NMBE5031551 and NMBE5031552), right hemi-mandible with i2 and p2-m3 (NMBE5026738, cast NMB-UM6719), reconstructed incomplete mandible with left and right dental series with p2-m3 (original specimen exposed in ML, cast NMBE5031541, cast NMB-AS76), right femur (NMBE5031542, cast NMB-UM6314), incomplete right tibia (NMBE5031543), right tibia (NMBE5031544, cast NMB-UM6315), right calcaneus (NMBE5031545), two right astragali (NMB-2017 and NMB-698), right MtII (NMBE5026812), right MtIII (NMBE5026811) from Wischberg (Switzerland, MN1).

Description

Skull. The skull NMBE5031538 (Figs. 4A–4C) is long and relatively narrow (Lcondyles-nasals = 575.5 mm, Lcondyles-premaxilla = 615.5 mm, Wfrontals = 158.5 mm), belonging to a large-sized adult rhinocerotid. It is incomplete and laterally compressed. It lacks the zygomatic arches, the occipital crest, as well as the anterior dentition and the right cheek tooth series, while only M1–3 are preserved in the left one. The dental remains are much worn, indicating an aged individual. The separated nasal bones are long, but less than the premaxilla, relatively thin and bear a lateral apophysis (Fig. 4A). Roughness for a small nasal horn is preserved at the tip of the nasals. In lateral view, the foramen infraorbitalis and the posterior border of the U-shaped nasal notch are both located above the P3, while the anterior border of the orbit is above the M1/2 limit. The minimum distance between the posterior edge of the nasal notch and the anterior border of the orbit is 82.5 mm. The back of the cheek teeth reaches the posterior half of the skull. The processus lacrymalis seems to be slightly developed and the processus postorbitalis of the frontal is absent. The base of the processus zygomaticus maxillari is high; it is about 2.5 cm above the neck of M2. The general dorsal profile of the skull is slightly concave, characterized by a nasal tip pointing downwards and by a slight posterior elevation of the parietal bones. In dorsal view, the postorbital constriction is very moderate, and the fronto-parietal crests are well-separated. The processus postglenoidalis is long, strong and transversally narrow. The articular surface of the latter defines a right dihedron in cross section. The processus postglenoidalis is curved forward and contacts the short processus-posttympanicus, partially closing the external auditory pseudomeatus. The processus paraoccipitalis is long and well developed (Fig. 4C). The foramen magnum is circular. A smooth median transverse ridge runs all over the occipital condyles, but there is no axial truncation.

Figure 4 Diaceratherium lemanense (Perissodactyla, Rhinocerotidae) from Wischberg locality, Bern Canton, Swiss Molasse basin (MN1, Agenian, earliest Miocene).

Skull NMBE5031538 in laterodorsal (A), ventral (B) and occipital (C) views. Right hemimandible NMBE5026738 in labial (D), lingual (E) and occlusal (F) views with an enlarged occlusal view of the teeth (G). Right maxillary fragment NMBE5031539 in labial (H), lingual (I) and occlusal (J) views with an enlarged occlusal view of the teeth (K). Photo credit: Patrick Röschli.

Mandible. The hemi-mandible NMBE5026738 (Figs. 4D–4F) bears a very weak median sagittal groove (sulcus mylohyoideus) on the lingual side of the corpus mandibulae (Fig. 4E). The symphysis is thick and not constricted at the diastema level in the preserved side (Fig. 4F). It is upraised about 30° with respect to the corpus mandibulae, and its posterior border, as well as the foramen mentale, is located below p2. The corpus mandibulae displays a straight ventral border with a constant height below p2–p4 (height below p2 = 80.3 mm) that gets slightly higher until m3 (height below m3 = 92.5 mm). The incisura vasorum is weakly marked, the angulus mandibulae not much developed and the retromolar space rather long. The foramen mandibulae (Fig. 4E) is located below the alveolar level. The other referred mandibular specimen (casts NMBE5031541 and NMB-AS76) is greatly reconstructed and the anterior part of the symphysis is missing. The ramus mandibulae (maximum height = 250.0 mm) is inclined forward, with a processus coronoideus sagittally well developed. The foramen mandibulae is also located much below the jugal teeth neck line.

Anterior teeth. The anterior dentition is reduced to the chisel-tusk shearing complex of I1-i2, characteristic of the family Rhinocerotidae sensu Radinsky (1966). The referred I1 are almond-shaped in cross section (Figs. 5A–5I) and the i2 is tusk-like (Figs. 5J–5L).

Figure 5 Diaceratherium lemanense (Perissodactyla, Rhinocerotidae) from Wischberg locality, Bern Canton, Swiss Molasse basin (MN1, Agenian, earliest Miocene).

Left I1 NMBE5031540 in occlusal (A), lingual (B) and labial (C) views. Right I1 NMBE5031546 in occlusal (D), lingual (E) and labial (F) views. Right I1 NMBE5031540 in occlusal (G), lingual (H) and labial (I) views. Left i2 NMBE5031547 in occlusal (J), lingual (K) and labial (L) views. Left P3 NMBE5031549 in occlusal (M) and lingual (N) views. Right P3 NMBE5031550 in occlusal (O) and lingual (P) views. Fragmentary right P1 NMBE5031548 in occlusal (Q), lingual (R) and labial (S) views. Fragmentary left p4 NMBE5031551 in occlusal (T), lingual (U) and labial (V) views. Photo credit: Patrick Röschli.

Upper cheek teeth. The cheek teeth are low-crowned (brachydont) and their roots are partly joined. There is neither cement nor secondary enamel foldings on the crowns of cheek teeth. The enamel is thin and wrinkled. Due to the advanced dental wear and the fragmented state of upper cheek teeth, only few characters can be identified. The protocone of upper molars and premolars is not constricted. The lingual and labial cingula are completely lacking on upper molars (Figs. 4B and 4H–4K), while the lingual one seems to be strong and continuous on P3 (NMBE5031549 and NMBE5031550, Figs. 5M–5P). The P1 NMBE5031548 is biradiculate and does not bear labial cingulum (Figs. 5Q–5S). The P3 is molariform (sensu Heissig, 1969), the paracone fold seems poorly developed on upper molars and the M3 is quadrangular in occlusal view (Figs. 4B and 4K), with a transverse protoloph and a posterior groove on the ectometaloph.

Lower cheek teeth. The lower dental formula is 1i-3p-3m (there are neither alveoli nor any trace of contact with the d1/p1 on p2). The lower premolars are not reduced compared with the molars (Lp3–4/Lm1–3 > 50; Figs. 4D–4G; Table 7). The lingual cingulum of the lower cheek teeth is reduced to the base of the opening of the anterior valley as an extension of the anterior cingulum (Fig. 4E). The labial cingulum is only present at the base of the paraconid, while the posterior is only present on lower premolars (Fig. 4D). The external groove is developed, oblique and vanishes before the neck. The trigonid is angular on lesser worn teeth and forms an acute dihedron with a rather developed lingual branch of the paralophid in occlusal view (Fig. 4G). The talonid valley is narrow and V-shaped on p2-m3. The p2 displays a developed paraconid and a constricted paralophid (spur-like), an open posterior valley, as well as marked anterior and external grooves of the ectolophid. The hypolophid is transverse on lower molars and the entoconid of the lower molars does not bear a lingual groove.

Table 7 Dimensions of the lower cheek teeth of Diaceratherium lemanense (Perissodactyla, Rhinocerotidae) from Wischberg locality, Bern Canton, Swiss Molasse basin (MN1, Agenian, earliest Miocene).

NMBE5026738	Casts NMBE5031541 and NMB-AS76	
Lower tooth row	Lp3–4	Lm1–3	Lp3–4/Lm1–3 × 100	Lower tooth row	Lp3–4	Lm1–3	Lp3–4/Lm1–3 × 100	
Right	78.0	137.0	56.9	Left	77.0	130.0	59.2	
				Right	76.5	133.5	57.3	
Lower cheek teeth	L	W		Lower cheek teeth	L	W	H	
Right p2	30.0	20.1		Left p2	28.5	–	24.2	
				Right p2	28.0	16.9	26.9	
Right p3	36.0	25.0		Left p3	38.2	22.1	–	
				Right p3	36.1	24.0	–	
Right p4	40.5	29.5		Left p4	36.5	29.0	–	
				Right p4	38.5	26.5		
Right m1	42.8	28.5		Left m1	39.5	28.7	–	
				Right m1	40.5	26.5		
Right m2	46.0	30.5		Left m2	44.2	30.5	27.5	
				Right m2	46.8	29.8	28.0	
Right m3	49.5	28.5		Left m3	47.5	28.5	31.0	
Note:

Dimensions are given in mm.

Femur. The femur (NMBE5031542, Figs. 6A–6D) is quite slender (Table 8) and anteroposteriorly compressed by deformation (Figs. 6B and 6D). The trochanter major is high, the articular facet of the head is slightly medially asymmetric, the fovea capitis is high and narrow, and the third trochanter is developed. In medial view (Fig. 6B), the anterior border of the diaphysis forms a slope break with the medial lip of the patellar trochlea. In anterior view (Fig. 6A), the distolateral epicondyle is low and well developed, the proximal border of the patellar trochlea is horizontal. The lateral lip is acute, while the lateral one is rounded.

Figure 6 Diaceratherium lemanense (Perissodactyla, Rhinocerotidae) from Wischberg locality, Bern Canton, Swiss Molasse basin (MN1, Agenian, earliest Miocene).

Right femur NMB-UM6314 in anterior (A), medial (B), posterior (C) and lateral (D) views. Right tibia NMBE5031544 in anterior (E), medial (F), posterior (G) and lateral (H) views. Photo credit: Patrick Röschli.

Table 8 Dimensions of the femur and tibia of Diaceratherium lemanense (Perissodactyla, Rhinocerotidae) from Wischberg and comparisons with other Diaceratherium and Pleuroceros specimens.

Long bones	L	TD
prox	APD
prox	TD
dia	APD
dia	TD
dist	APD
dist	
Femur								
D. lemanense
Wischberg (this study)	499.0	187.5	69.0	66.0	55.0	132.0	130.5	
D. lemanense
Gannat (Type, coll. MNHN)	(465.0)	–	–	(106.0)	–	160	>142	
D. lamilloquense
Castelmaurou (Duranthon, 1990)	–	–	–	53.6	49.0	–	120.0	
D. tomerdingense
Tomerdingen (Type, coll. SMNS)	–	–	–	–	–	123.8	151.9	
D. asphaltense
Pyrimont-Challonges (Type, Depéret & Douxami, 1902)	429.0	168.0	–	63.0	–	117.0	–	
D. aginense
Laugnac (coll. MHNM)	490.0	165.0	90.0	61.6	52.0	125.0	157.0	
D. aurelianense
Neuville-aux-Bois (Cerdeño, 1993)	433.0	161.0	84.9	59.0	43.5	129.0	140.0	
Tibia								
D. lemanense
Wischberg (this study)	380	124.5	84.4	65.0	40.5	93.8	48.3	
D. lemanense
Gannat (Type, coll. MNHN)	381	132	99	56	–	103	–	
P. blandfordi
Bugti Hills (Antoine et al., 2010)	–	–	–	47.0–47.0
47.0 [2]	33.5–36.0
34.5 [3]	73–78.5
76.1 [4]	(50.0)–57.5
52.9 [4]	
D. lamilloquense
La Milloque (Michel, 1983)	–	114.0	80.0	–	–	–	–	
D. lamilloquense
Castelmaurou (Duranthon, 1990)	337.0	109.0	77.5	45.0	46.5	78.0	70.0	
D. tomerdingense
Tomerdingen (Type, coll. SMNS)	–	94.8	–	52.0	41.8	–	–	
D. asphaltense
Pyrimont-Challonges (Type, Depéret & Douxami, 1902)	(351.0)	111.0	–	45.0	–	90.0	–	
D. asphaltense
Saulcet (coll. NMB)	361.0	123.8	109.5	56.0	46.5	98.6	64.0	
D. aginense
Laugnac (coll. MHNM)	322.0–360.0
338.0 [4]	113.0–128.0
120.5 [3]	95.5–102.0
98.4 [4]	45.3–49.9
48.3 [4]	39.6–48.6
43.9 [4]	87.2–92.1
89.0 [4]	55.5–61.9
60.0 [4]	
D. aurelianense
Neuville-aux-Bois (Cerdeño, 1993)	274.0–288.0
281.0 [2]	102.0–112.0
107.0 [2]	104.0	46.0–51.0
48.5 [2]	40.3–47.0
43.7 [2]	94.4	60.0–68.2
64.1 [2]	
Note:

Dimensions are given in mm, those in parentheses are estimated and those in italics are based on the literature. Localities are indicated below the taxon name and those in bold font are the type localities of the species. The upper line indicates the minimum and maximum dimensions, and the bottom line (when several specimen are used) indicates the average value and the number of specimens (in brackets).

Tibia. Two tibiae are preserved: the specimen NMBE5031544 is complete (Figs. 6E–6H), well preserved and anteroposteriorly compressed by deformation, while the other (NMBE5031543) is incomplete. In distal view, the anterodistal groove is well marked. The mediodistal gutter for the m. tibialis is present and shallow, and the posterior apophysis is high and rounded (Fig. 6G). In lateral view (Fig. 6H), the proximal articulation for the fibula is low and the diaphysis bears discontinuous contact marks for the fibula.

Astragalus. Two astragali are preserved (Figs. 7A–7F). They slightly differ by their dimensions, but they are proportionally and morphologically homogeneous (Table 9). The fibula-facet is subvertical and transversely flat (Figs. 7A and 7D). The collum tali is high. In proximal view, the posteroproximal border of the trochlea is sinuous. In distal view (Figs. 6C and 6F), the trochlea is very oblique compared to the distal articulation and the posterior stop on the cuboid-facet is present on NMB-2017 (not observable in NMB-698). The lateral lip is very prominent (Figs. 7A and 7D), and the medial tubercle is low, salient and laterally displaced. The calcaneus-facet 1 (sensu Heissig, 1972) is very concave. The laterodistal expansion of this facet is broken, but was probably short (Figs. 7B and 7E). The calcaneus-facet 2 is roughly oval, flat and wider than high. The calcaneus-facet 3 is transversely developed and separated from the calcaneus-facet 2 by a notch (Figs. 7B and 7E).

Figure 7 Diaceratherium lemanense (Perissodactyla, Rhinocerotidae) from Wischberg locality, Bern Canton, Swiss Molasse basin (MN1, Agenian, earliest Miocene).

Right astragalus NMB-2017 in anterior (A) posterior (B) and distal (C) views. Right astragalus NMB-698 in anterior (D) posterior (E) and distal (F) views. Right calcaneus NMBE5031545 in distal (G), anterior (H), lateral (I), posterior (J) and medial (K) views. Right MtIII NMBE5026811 in anterior (L), lateral (M), posterior (N), medial (O) and proximal with dorsal toward top (P) views. Right MtII NMBE5026812 in proximal with dorsal toward top (Q), anterior (R), lateral (S), posterior (T) and medial (U) views. Photo credit: Patrick Röschli.

Table 9 Dimensions of the astragalus and calcaneum of Diaceratherium lemanense (Perissodactyla, Rhinocerotidae) from Wischberg and comparisons with other Diaceratherium and Pleuroceros specimens.

Tarsals	TD	APD	H	
Astragal				
D. lemanense
Wischberg (this study)	76.8–85.5
81.2 [2]	40.0–41.5
40.8 [2]	70.5–74.0
72.3 [2]	
D. lemanense
Gannat (coll. NMB)	87.1	65.0	70.3	
P. pleuroceros
Paulhiac (De Bonis, 1973)	(67.5)	–	(62.25)	
P. blandfordi
Bugti Hills (Antoine et al., 2010)	71.5–75.5
73.6 [5]	47.5–49.5
48.1 [5]	(58.0)–64.5
62.5 [5]	
D. lamilloquense
Castelmaurou (Duranthon, 1990)	76.7–78.5
77.6 [2]	54.7–63.4
59.05 [2]	69.5–73.5
71.5 [2]	
D. asphaltense
Saulcet (coll. NMB)	90.0	62.0	86.0	
D. aginense
Laugnac (coll. MHNM)	83.0–85.6
84.5 [3]	50.0–55.1
51.9 [3]	70.7–72.1
71.5 [3]	
D. aurelianense
Neuville-aux-Bois (Cerdeño, 1993)	75.5–86.0
81.4 [3]	72.6–77.0
74.8 [2]	61.2–68.4
63.8 [3]	
D. askazansorense
Askazansor (Kordikova, 2001)	85.0	49.0	73.8	
Calcaneum				
D. lemanense
Wischberg (this study)	–	65.5	124.4	
P. pleuroceros
Paulhiac (De Bonis, 1973)	–	(54.8)	(100.5)	
P. blandfordi
Bugti Hills (Antoine et al., 2010)	(67)	55.0–63.0
59.7 [3]	(97.0)–105.0
104.0 [2]	
D. lamilloquense
La Milloque (Michel, 1983)	–	60.5	–	
D. lamilloquense
Castelmaurou (Duranthon, 1990)	73.0	50.5–55.5
53.0 [2]	107.0–115.0
111.0 [2]	
D. asphaltense
Pyrimont-Challonges (Type, coll. UCBL)	69.1	63.1	108.3	
D. asphaltense
Saulcet (coll. NMB)	77.5	62.5	120.0	
D. aginense
Laugnac (coll. MHNM)	72.2	55.0–61.0
58.5 [4]	104.5–115.2
110.7 [4]	
D. aurelianense
Neuville-aux-Bois (Cerdeño, 1993)	71.6–75.8
73.7 [2]	50.5–62.5
56.5 [2]	122.0–123.0
122.5 [2]	
D. askazansorense
Askazansor (Kordikova, 2001)	80.5	63.0	118.5	
Note:

Dimensions are given in mm, those in parentheses are estimated and those in italics are based on the literature. Localities are indicated below the taxon name and those in bold font are the type localities of the species. The upper line indicates the minimum and maximum dimensions, and the bottom line (when several specimen are used) indicates the average value and the number of specimens (in brackets).

Calcaneum. The calcaneum NMBE5031545 (TDmax = -, APDmax = 65.5 mm, H = 124.4 mm; Figs. 7G–7K) is incomplete, the sustentaculum tali is not preserved. Both fibular and tibial facets are lacking. The tuber calcanei is high and slender in distal view (Fig. 7G). The insertion for the m. fibularis longus is marked on the lateral side (Fig. 7I), forming a deep notch. The corpus calcanei is long (APD = 51.5) and narrow (TD = 27.2). The cuboid-facet forms a transverse half-circle in distal view, and it is slightly convex anteroposteriorly.

Metatarsals. The metatarsals have a long insertion for the m. interossei (Figs. 7M, 7O and 7S), a salient insertion for the m. extensor carpalis (Figs. 7L and 7R), and a high and acute median keel of the distal articulation. The MtII bears a narrow and sagittally-elongated proximal end (Fig. 7Q). The mesocuneiform facet forms a half oval. An axially elongated posteromedial entocuneiform-facet joins the proximal facet (Fig. 7T). On the lateral side (Fig. 7S), the posterior ectocuneiform facet is large and lozenge-shaped while the anterior one is smaller and nearly vertical. They are separated by a large groove. The anterior and posterior MtIII-facets are poorly developed and connected to the anterior and posterior ectocuneiform facets, respectively. The cuboid-facet of the MtIII NMBE5026811 is absent. In proximal view (Fig. 7P), the anterior border of the MtIII is slightly curved, while it is concave and high laterally, in anterior view (Fig. 7L). The MtIV-facets are independent (Fig. 7M), the posterior one is distally displaced with respect to the anterior one. The diaphysis slightly widens distally, reaching its maximal width immediately above the distal articulation, especially due to the epicondyles. No posterodistal tubercle is present on the diaphysis (Fig. 7N). The MtIII NMBE5026811 is rather robust (GI = 30.8; Table 10), while the MtII NMBE5026812 is shorter and more slender (GI = 24.6; Table 10).

Table 10 Dimensions of the MtII and MtIII of Diaceratherium lemanense (Perissodactyla, Rhinocerotidae) from Wischberg and comparisons with other Diaceratherium and Pleuroceros specimens.

Metatarsals	L	TD
prox	APD
prox	TD
dia	APD
dia	TD
dist	APD
dist	
MtII								
D. lemanense
Wischberg (this study)	130.6	31.7	–	32.1	16.3	36.2	29.5	
D. lemanense
Gannat (coll. NMB)	134.0	30.0	42.0	27.0	24.0	41.0	37.5	
P. pleuroceros
Paulhiac (De Bonis, 1973)	(111.0)	(21.8)	(28.5)	(18.0)	(15.0)	(30.0)	–	
P. blandfordi
Bugti Hills (Antoine et al., 2010)	101.5	22.5	34.5	21.0	19.0	28.0–30.0
29.0 [2]	28.5–29.0
28.75 [2]	
D. lamilloquense
Castelmaurou (Duranthon, 1990)	131.5–132.2
131.9 [2]	27.0–27.0
27.0 [2]	34.0–35.4
34.7 [2]	22.0–24.0
23.0 [2]	19.0–20.5
19.8 [2]	33.4–34.0
34.7 [2]	33.4–35.3
34.4 [2]	
D. asphaltense
Saulcet (coll. NMB)	119.5	42.0	41.8	26.8	22.5	30.5	39.1	
D. aginense
Laugnac (coll. MHNM)	104.9–113.4
108.9 [5]	29.0–31.2
29.8 [6]	34.3–37.3
36.1 [5]	24.4–27.6
26.0 [6]	19.0–22.3
20.5 [6]	33.0–35.1
33.7 [4]	35.6–38.9
36.9 [4]	
D. aurelianense
Neuville-aux-Bois (Cerdeño, 1993)	101.4	35.7	37.5	30.4	20.5	41.6	42.7	
MtIII								
D. lemanense
Wischberg (this study)	146.9	47.4	35.5	45.4	16.3	47.3	30.6	
D. lemanense
Gannat (coll. NMB)	153.0	54.5	44.0	42.0	20.0	59.0	40.0	
P. pleuroceros
Paulhiac (De Bonis, 1973)	–	(42.0)	(40.5)	(37.5)	–	(45.0)	–	
P. blandfordi
Bugti Hills (Antoine et al., 2010)	–	41.0	(34.0)	31.5–33.0
32.8 [2]	15.5–16.0
15.8 [2]	36.0–36.0
36.0 [2]	30.0–32.5
31.3 [2]	
D. lamilloquense
Castelmaurou (Duranthon, 1990)	141.5–144.0
142.3 [2]	44.0–48.2
46.1 [2]	34.0–36.4
35.2 [2]	34.9–38.5
36.7 [2]	17.2–18.3
17.8 [2]	51.2–43.0
47.1 [2]	36.3–39.4
37.9 [2]	
D. asphaltense
Pyrimont-Challonges (Type, coll. UCBL)	128.0	(39.0)	42.5	39.4	18.0	51.1	33.5	
D. asphaltense
Saulcet (coll NMB)	131.5	50.6	45.7	40.0	23.0	49.3	44.1	
D. aginense
Laugnac (coll. MHNM)	122.5–131.3
127.5 [6]	41.4–48.5
45.0 [5]	38.2–44.8
41.3 [5]	37.6–45.3
41.7 [6]	18.2–21.0
19.3 [6]	47.3–54.1
51.1 [6]	39.6–42.1
40.5 [6]	
D. aurelianense
Neuville-aux-Bois (Cerdeño, 1993)	117.0–118.0
117.5 [2]	48.3–50.7
49.8 [2]	42.8	40.5–43.8
42.1 [3]	17.3–17.8
17.6 [2]	53.2–54.0
53.9 [3]	39.0–43.3
41.2 [2]	
Note:

Dimensions are given in mm, those in parentheses are estimated and those in italics are based on the literature. Localities are indicated below the taxon name and those in bold font are the type localities of the species. The upper line indicates the minimum and maximum dimensions, and the bottom line (when several specimen are used) indicates the average value and the number of specimens (in brackets).

Remarks

Based on dimensions and morphology, the referred specimens cannot be assigned to the small-sized contemporaneous European rhinocerotids. They differ from Protaceratherium minutum by larger dimensions, a thick mandibular symphysis, a reduced labial and lingual cingulum on lower cheek teeth, an astragalus wider than high and the separation between Cc1 and Cc2 facets (De Bonis, 1973; Ginsburg, Huin & Locher, 1981). They also differ from Pleuroceros pleuroceros by larger dimensions, as well as a developed external groove on lower cheek teeth, a reduced lingual cingulum on lower premolars, a subvertical fibula facet of the astragalus and an MtIII with a curved and oblique proximal border in anterior view (De Bonis, 1973; Antoine et al., 2010). Plesiaceratherium Young, 1937 and Mesaceratherium are roughly of similar size. The referred material differs from Plesiaceratherium by a developed ectolophid groove on lower cheek teeth, the absence of external roughnesses on p2–3, an ascending ramus of the mandible not inclined backwards, and much stouter metapodials (Yan & Heissig, 1986; D. Becker, 2018, personal observation). It differs from Mesaceratherium by a less strongly raised symphysis, an astragalus wider than high and an anteroposteriorly reduced proximal facet of the MtIII (Heissig, 1969; De Bonis, 1973).

The assignment of the referred specimens to the genus Diaceratherium is supported by their dimensions and morphology. The nasals (long, thin and totally separated), the deep, U-shaped notch ending above P3, the orbital features (presence of a processus lacrymalis, anterior border above M1/2), the mandible (straight profile of the base of the corpus mandibulae), the dental remains (quadrangular M3, constricted paralophid and developed paraconid on p2) and the astragali (lateral lip larger than the medial one and a low, salient and laterally displaced medial tubercle) present the characteristics of the genus Diaceratherium (Becker et al., 2009, 2010, 2018; Antoine et al., 2010; C. Jame, J. Tissier, D. Becker, 2019, personal observation). However, the attribution of the studied material to a species within this genus remains difficult. Apart from D. massiliae Ménouret & Guérin, 2009, whose generic attribution remains doubtful by several non-Diaceratherium morphological features (Antoine & Becker, 2013), between five and seven species of this genus are usually considered as valid in the literature (Heissig, 1999; Boada-Saña, Hervet & Antoine, 2007; Becker et al., 2009; Antoine & Becker, 2013).

The studied material differs from the type species D. tomerdingense by the absence of lingual cingulum under the protocone and at the opening of the median valley on M1–2, as well as an oblique external groove of lower premolars in labial view that does not vanish before the neck (Dietrich, 1931; D. Becker and J. Tissier, 2019, personal observation). Furthermore, though the metatarsals cannot be directly compared to the metacarpals from Tomerdingen, they would be much longer, because the metacarpals of D. tomerdingense are close to those of D. aginense from Laugnac, whose metatarsals are shorter than those from Wischberg (Table 4). However, the nasal bones are rather similar in size and shape, and also bear a small rugosity for the horn.

The specimens from Wischberg differ from the latest Oligocene diacerathere, D. lamilloquense, from La Milloque by the absence of lingual cingulum under the protocone of M3, a less angular trigonid on lower cheek teeth, and a low proximal articulation for the fibula on the tibia (Michel, 1983; Brunet, De Bonis & Michel, 1987). They differ from the specimens from Castelmaurou by the absence of labial cingulum in the external groove of m2 and m3, and the absence of a posterior facet for the MtII on the MtIII (Duranthon, 1990).

The skull NMBE5031538 and mandible NMBE5026738 differ from the type material of D. asphaltense from Pyrimont in having slightly stouter and shorter nasals, a moderate postorbital constriction of the skull, more distant frontoparietal crests, as well as a higher corpus of the mandible and a lower position of the foramen mandibulae on the ramus (Depéret & Douxami, 1902; D. Becker and J. Tissier, 2018, personal observation). Concerning the postcranial remains, some differences can be noted with D. asphaltense from Pyrimont and Saulcet, such as an anteroposteriorly reduced proximal facet of the MtIII for the ectocuneiform, a laterally compressed distal facet of the calcaneus for the cuboid and a slender tuber calcanei (Depéret & Douxami, 1902; D. Becker and J. Tissier, 2018, personal observation).

The material from Wischberg differs from D. aginense from Laugnac (type locality) in displaying a partially closed external auditory pseudomeatus, a less developed ectolophid groove of lower cheek teeth, a weaker lingual groove on the corpus mandibulae, a longer posterodistal apophysis of the tibia, more gracile metapodials and a straighter navicular facet of the astragalus in anterior view (Répelin, 1917; D. Becker and J. Tissier, 2018, personal observation).

It differs from D. aurelianense by the absence of labial cingulum on lower molars, the absence of the postorbital process of the frontals, the weaker lesser trochanter of the femur and the more gracile and longer metapodials, with an acute median keel in distal view (Mayet, 1908; Cerdeño, 1993; D. Becker and J. Tissier, 2018, personal observation).

Finally, it differs from the Early Miocene Kazakh species D. askazansorense by a smaller size of the lower molars, a constant height of the horizontal ramus, less hypsodont teeth, a lower collum tali of the astragalus and a slender tuber calcanei (Kordikova, 2001).

The cranio-dental and postcranial characters of the diacerathere from Wischberg are in fact morphologically indistinguishable from those of D. lemanense from Gannat (type locality). The nasals are small, and same sized as the type skull from Gannat. Like the specimen from Gannat NMB Gn. 40, the proximal facet of the MtIII is sagitally elongated and concave in anterior view. The astragalus from this same individual is very similar to the two specimens from Wischberg and is also wider than high. As in D. lemanense from Montaigu (NMB S.G.18480), the ramus mandibulae is inclined forward, with a sagittally well developed processus coronoideus. The lingual and labial cingula are also absent on lower cheek teeth. The material from Wischberg only differs by a slightly smaller size than the type material. Therefore, we attribute the referred specimens from Wischberg to D. lemanense.

Discussion

Systematic remarks

The systematics of the genus Diaceratherium is far from consensual. Four species in particular are contentious and often subject to synonymies: D. lemanense, D. asphaltense, D. tomerdingense and D. aginense.

After Antoine & Becker (2013) and Becker et al. (2018), D. tomerdingense is a junior synonym of D. aginense and the latter is likely to be a junior synonym of D. asphaltense. More recently, Becker et al. (2018) still accepted the synonymy of D. tomerdingense and D. aginense, but maintained D. asphaltense as valid whereas, according to De Bonis (1973) and Boada-Saña, Hervet & Antoine (2007), D. asphaltense and D. tomerdingense should be considered as junior synonyms of D. lemanense. However, no clear justification is ever provided, except for the synonymy of D. asphaltense and D. lemanense by the phylogenetic analysis of Boada-Saña (2008). Yet, the coding of D. asphaltense in this work is based on photographs of the type material from Pyrimont-Challonges (Boada-Saña, 2008: tab. 1). As a matter of fact, based on direct examination of the type material of D. lemanense, D. asphaltense, D. aginense and D. tomerdingense, we conclude that these four species can be differentiated based on their morphology, and can be considered as valid, as expressed in the emended diagnosis of D. lemanense.

These synonymies probably derived from the absence of differential diagnoses between these four species, and of designated type for D. lemanense. Indeed, a skull referred to “Acerotherium” lemanense from the type locality of Gannat (Roman, 1912, Pl. VIII fig. 1-3) was unfortunately mistakenly considered as a reference specimen for comparison by Becker et al. (2009, 2018) whereas Boada-Saña, Hervet & Antoine (2007) had designated another skull and mandible from Gannat (MNHN AC 2375 and MNHN AC 2376, respectively) as lectotype. Regrettably, both skulls from Gannat may belong to two different taxa, which led to unfortunate comparisons of specimens and erroneous taxonomic attributions. The skull used by Becker et al. (2009, 2018) as reference material of D. lemanense (FSL-213944) is remarkably similar to the skull attributed to D. lemanense from Eschenbach (NMSG–P2006/1), but after direct observation could both be referred to Plesiaceratherium. Moreover, cranial remains from Saulcet (NMB-SAU-1662) and Bühler (NMSG-F13607) have been referred to D. asphaltense (Becker et al., 2009, 2018), based on similarities with the type skull of D. asphaltense from Pyrimont-Challonges (FSL-212997bis), but also on indisputable dissimilarities with the non-Diaceratherium skulls from Gannat (FSL-213944) and Eschenbach (NMSG–P2006/1).

Finally, another systematic interpretation has been recently proposed by Heissig (2017), who referred the species D. aurelianense to the genus Prosantorhinus because of characters not found in other species of the genus Diaceratherium. These characters are “the deeply concave skull profile with upslanting nasals, a wide nasal incision of medium depth, and the triangular last upper molar.” Similarities between the two genera had already been expressed by Cerdeño (1996) who referred some specimens previously attributed to D. aurelianense to the genus Prosantorhinus but keeping both taxa as valid. Antoine et al. (2018) have also recently attributed all the material previously referred as D. aurelianense from Béon 2 to Prosantorhinus aff. douvillei, which indicates indeed similarities between these two taxa, as also already noted by Mayet (1908). However, Antoine et al. (2018) subsequently expressed numerous anatomical differences between these two taxa, including the 20% size difference of the MtIV, which is a character that specifically distinguishes these two genera. Moreover, the characters used by Heissig (2017) seem quite labile to confirm the attribution of the species D. aurelianense to the genus Prosantorhinus. Indeed, a recently described skull of D. asphaltense from Bühler does show a deeply concave skull and slightly upslanted nasals (Becker et al., 2018), though not as much as the skull of D. aurelianense (Fig. 8). Another skull of D. asphaltense from Saulcet has a similar morphology, but it is true that D. lemanense and D. aginense do not show such an upslanted nasal bone (though for the latter species the skulls illustrated by Répelin (1917) are heavily reconstructed, and the global shape is very misleading). Finally, the M3 is indeed more triangular in D. aurelianense than in other species of the genus, but it could be a character specific to this species. Therefore, to the best of our knowledge, the four above-mentioned problematic Diaceratherium species should be considered as valid, just as D. lamilloquense and D. askazansorense, and D. aurelianense should still belong to the genus Diaceratherium (as presented in Table 11).

Figure 8 Comparison of the skulls of Diaceratherium (Perissodactyla, Rhinocerotidae).

(A) D. asphaltense (NMSG-F13607) from Bühler (MP30-MN1; Becker et al., 2018). (B) D. apshaltense (NMB Sau 1662) from Saulcet (MN1). (C) D. aurelianense (MHNT.PAL.2013.0.1001, cast of the holotype) from Neuville-aux-Bois (MN3). (D) D. aginense (FSL collection) from Laugnac (MN2). (E) D. lemanense (MNHN-AC-2375, holotype) from Gannat (MN1). (F) D. lemanense (cast NMBE5031538) from Wischberg (MN1).

Table 11 Occurrences of Diaceratherium (Perissodactyla, Rhinocerotidae) species in France, Switzerland and other countries.

P-MN zones	Taxa	Localities	
France	Switzerland	Others	
MN4	D. aurelianense	Artenay		Areeiro da Barbuda (Portugal), Areeiro de Santa Luzia (Portugal), Eggingen-Mittelhart 3 (= D. cf. aurelianense; Germany), Quinta da Carrapata (Portugal), Quinta da Noiva (Portugal), Quinta da Trindade (Portugal), Quinta das Pedreiras (Portugal), Quinta do Narigão (Portugal), Vale Pequeno (Portugal)	
MN3	D. aurelianense	Neuville-aux-Bois, Beaulieu, Chilleurs-aux-Bois, Chitenay, Esvres, La Brosse, Les Beilleaux, Les Buissonneaux, Marsolan, Mauvières, Navère, Ronville	Brüttelen, Cheyres, La Molière	Horta das Tripas (= D. cf. aurelianense; Portugal), Molí Calopa (Spain), Rubielos de Mora (Spain), Wintershof-West (Germany)	
MN2/3	D. askazansorense			Askazansor (Kazakhstan)	
MN2	D. aginense	Laugnac, Auterive, Beaupuy, Calmont-St-Cernin, Cintegabelle, Grépiac, Montaigu-le-Blin, Pouvourville, Venerque	Engehalde, La Chaux, Lausanne, Sous-le-Mont	Hessler (Germany)	
D. aurelianense			Loranca del Campo (= D. cf. aurelianense; Spain)	
D. lemanense	Barbotan-les-Thermes, Cindré, Gans, Laugnac, Montaigu-le-Blin, Selles-sur-Cher, St-Gérand-le-Puy	Engehalde	Budenheim (Germany), Ulm-Michelsberg (Germany)	
MN1	D. aginense	Gannat, Paulhiac			
D. asphaltense	Pyrimont-Challonges, Saulcet			
D. lemanense	Gannat, Bazas, Bézac, Caignac, Casteljaloux-Balade, Cindré, Ginestous, Grenade-sur-Garonne, Labastide-Beauvoir, Pechbonnieu, La Roche-Blanche-Gergovie, Paulhiac, Pech David, Randan, St-Loup Cammas, St-Michel-du-Touch, Saulcet, Saverdun, Toulouse Borderouge, Toulouse Embouchure	Wischberg	Finthen (Germany), Oppenheim (Germany), Weisenau (Germany)	
D. tomerdingense			Tomerdingen (Germany)	
MP30/MN1	D. asphaltense		Bühler		
MP30	D. lemanense	Billy, Gannat «sommet», Thézels (= D. aff. lemanense), Toulouse-Borderouge		Rott bei Bonn (Germany)	
MP29	D. lamilloquense	La Milloque, Castelmaurou, Castelnau d’Estretefonds, Dieupentale	Rickenbach		
Note:

Modified from Becker et al. (2009) with additions from Duranthon (1990, 1991), Antoine, Duranthon & Tassy (1997), Boada-Saña, Hervet & Antoine (2007), Mennecart et al. (2012), Antoine & Becker (2013) and Becker et al. (2018). Names in bold font indicate the type locality of the species.

Paleoecology and diversification

The Agenian rhinocerotid fauna from Wischberg includes two co-occurring mediportal species: the large-sized single-horned D. lemanense, and the small-sized tandem-horned Pleuroceros pleuroceros. The two taxa also differ by their body masses (Table 12), the former being a megaherbivorous with a body mass estimated to be over 103 kg (Owen-Smith, 1988). The record of Pleuroceros pleuroceros and D. lemanense is typical from the MN1 biozone, Gannat (France) being the type locality of both taxa, and is comparable to some contemporaneous French localities such as Paulhiac, Pyrimont-Challonges and Saulcet. This sympatric association is characteristic of the MN1 biozone and results from the faunal renewal starting at MP28 in Western Europe (Scherler et al., 2013). The presence of D. lemanense in Wischberg extends the record of this genus in Switzerland. Indeed, though the species D. lemanense was found in numerous French localities, Wischberg is the only record of this species in Switzerland during the MN1 biozone (Table 11). The genus Diaceratherium has a rather long record in Europe, from the late Oligocene to the early middle Miocene, and it survives the Oligocene–Miocene transition and diversifies then, with the presence of four different species during MN1: D. tomerdingense (type species), D. lemanense, D. asphaltense and D. aginense. In addition, Diaceratherium and Pleuroceros are also found in Asia at the same period: Pleuroceros blanfordi in Pakistan (Antoine et al., 2010) and D. askazansorense in Kazakhstan (Kordikova, 2001), which could indicate rhinocerotid exchanges between Europe and Asia.

Table 12 Estimates of rhinocerotid species body mass from Wischberg locality, Bern Canton, Swiss Molasse basin (MN1, Agenian, earliest Miocene).

Specimen/source	Estimated body mass (kg)	
Legendre (1989)	Mean L m1	Mean W m1		
Diaceratherium lemanense
NMBE5026738	42.8	28.5	1,730	
Diaceratherium lemanense
casts NMBE5031541 and NMB-AS76	40.5	26.5	1,417	
Pleuroceros pleuroceros
casts NMBE5031553 and NMB-AS77	29.7	18.6	504	
Tsubamoto (2014)	Li1		
Diaceratherium lemanense
Astragalus NMB 698	65.7	937	
Diaceratherium lemanense
Astragalus NMB 2017	69.7	1,105	
Fortelius & Kappelman (1993)	F1		
Diaceratherium lemanense
Femur NMBE5031542	500	1,624	
Fortelius & Kappelman (1993)	F5		
Diaceratherium lemanense
Femur NMBE5031542	132	1,261	
Fortelius & Kappelman (1993)	T2		
Diaceratherium lemanense
Tibia NMBE5031544	123	1,365	
Fortelius & Kappelman (1993)	T4		
Diaceratherium lemanense
Tibia NMBE5031544	88	1,104	
Fortelius & Kappelman (1993)	T5		
Diaceratherium lemanense
Tibia NMBE5031544	53	715	
Note:

Based on the allometric correlations with the occlusal surface of the first lower molar (Legendre, 1989), the transverse width of the tibial trochlea of the astragalus (Li1; Tsubamoto, 2014) and various femoral and tibial measurements (F1, F5, T2, T4, T5; Fortelius & Kappelman, 1993).

The Agenian period is marked by the beginning of a major worldwide diversification phase of Rhinocerotidae that lasted until the Late Miocene (Cerdeño, 1998), and during which perissodactyls reached the maximum body size and mass among terrestrial mammals (Smith et al., 2010). This rhinocerotid diversification may be due to the extinction of other megaherbivorous competitors in Europe such as the Anthracotheriinae (latest Oligocene, Scherler, 2011; Scherler, Lihoreau & Becker, 2018) or the Amynodontidae (Late Oligocene, Malez & Thenius, 1985). As for the other European perissodactyls, except for the Tapiridae, which are present in Europe until MN4, Paleotheriidae are extinct since MP25 (Rémy, 1995), Chalicotheriidae only re-appear during MN2 (Coombs, 2009), Equidae first appear with Anchitherium in MN3 (Kaiser, 2009; Alberdi & Rodríguez, 2012) and Eggysodontidae disappear in MN1 (Scherler et al., 2013). However, none of those reached sizes over 103 kg during this time. Within the Artiodactyla only nine genera were present in Europe during MN1 (Scherler et al., 2013) and all of them were smaller than the smallest rhinocerotids (Scherler, 2011; Mennecart, 2012). Finally, the proboscideans, another group of megaherbivores that will later dominate the megaherbivore communities, do not appear in Europe until mid-Orleanian times (MN3b; Antoine, Duranthon & Tassy, 1997; Göhlich, 1999). As a result, the earliest Miocene is a period during which rhinocerotids are the dominating largest herbivores and the only megaherbivores in Europe (Rössner & Heissig, 1999; Scherler et al., 2013). This observation is of particular interest since, like extant African megaherbivores, Early Miocene rhinocerotids likely had large food intake requirements and could have been able to subsist on low-quality (i.e., high fiber) food resources (Demment & Van Soest, 1985; Owen-Smith, 1988; Illius & Gordon, 1992). Furthermore, due to their size, Early Miocene megaherbivorous rhinocerotids are expected, like extant ones, to display specific life-history attributes, physiology and ecological characteristics related to their body mass (Blueweiss et al., 1978; Brown et al., 2004), such as larger geographic ranges, higher potential for dispersal (Brown, 1995; Gaston, 2003), lower mortality rates and better resistance to limiting environmental factors (Erb, Boyce & Stenseth, 2001). As a result, megaherbivores are considered to be a separate trophic guild among large herbivores (Fritz et al., 2002), possibly better adapted to ecosystems with high plant biomass but low-quality vegetation (Bell, 1982).

The beginning of the Miocene is marked by a short glacial event (Mi-1; Zachos et al., 2001). This sudden climatic event induced significant changes in the European vegetation. We observe indeed a lower proportion of C4 plants during the MN1 than during the Oligocene (Urban et al., 2010) and an increase of mesothermic vegetations at the expense of megathermic ones (Mosbrugger, Utescher & Dilcher, 2005; Bessedik, Guinet & Suc, 1984) which may have promoted fiber-rich plants associations at the beginning of the Miocene (e.g., Leguminosae; Berger, 1990). Janis (1976) hypothesized that perissodactyls (hindgut fermenters) were able to overcome competition of other large herbivorous mammals, especially ruminants (foregut fermenters), by their ability to tolerate more fibrous herbage. This could explain the diversification of rhinocerotids at the beginning of the Miocene, for which large size and digestive system might have increased their ability to monopolize resources (Fritz et al., 2002) and extract nutrients from specific feeding niches (Illius & Gordon, 1992). The evolutionary success and rapid diversification of rhinocerotids during the earliest Miocene could consequently be linked to this particular environmental change, triggered by the short glaciation event but also by the absence of other megaherbivores. After the late Oligocene faunal renewal (Scherler et al., 2013), the earliest Miocene, and especially the first 1 million-year period (MN1), may have been a crucial time period for the Rhinocerotidae, and especially megaherbivorous taxa, to start diversifying by occupying new ecological niches available at that time. Further analyses taking into account all European rhinocerotids, with their masses and anatomical features, will be necessary to test this hypothesis and better understand this unique transition in the European assemblages of megaherbivores at the beginning of the Miocene.

Conclusions

Based on comparisons, the rhinocerotid specimens from Wischberg, a typical early Agenian (MN1) locality, can be attributed to two different taxa: D. lemanense and Pleuroceros pleuroceros. Though Schaub & Hürzeler (1948) had identified a third taxon, D. asphaltense, we believe that the material identified as such should be attributed to the coeval D. lemanense, based on morphological differences with the holotype of D. asphaltense from Pyrimont-Challonges (MN1, France). Furthermore, we believe that all the species of Diaceratherium found at the present time in the literature could be considered as valid, until an extensive revision of this genus is performed, preferentially through a phylogenetic analysis.

We are greatly indebted to all curators of the collections visited during this work, who kindly helped us during our visit: Manuela Aiglstorfer and Reinhard Ziegler (SMNS), Christine Argot and Guillaume Billet (MNHN), Christophe Borrely (MHNM), Jana Fehrensen (ML), Yves Laurent (MHNT), Ursula Menkveld-Gfeller (NMBE), Emmanuel Robert (FSL) and especially Loïc Costeur (NMB) who kindly provided additional measurements. We thank Clara Stefen for her editorial work. Finally, we would like to gratefully acknowledge Luca Pandolfi, Esperanza Cerdeño and Pierre-Olivier Antoine for their insightful reviews and comments, which helped us to greatly improve our draft.

Abbreviations

APD antero-posterior diameter

Cc calcaneus

dia diaphysis

dist distal

FSL Faculté des Sciences de l’Université de Lyon

GI gracility index

H height

I/i upper/lower incisor

L length

M/m upper/lower molar

Mc metacarpal

MHNM Muséum d’histoire naturelle de Marseille

MHNT Museum d’histoire naturelle de Toulouse

ML Museum Langenthal

MNHN Muséum national d’Histoire naturelle (Paris)

Mt metatarsal

NMB Naturhistorisches Museum Basel

NMBE Naturhistorisches Museum der Burgergemeinde Bern

P/p upper/lower premolar

prox proximal

SMNS Staatliches Museum für Naturkunde Stuttgart

TD transversal diameter

W width.

Additional Information and Declarations

Competing Interests

Author Contributions

Data Availability

The authors declare that they have no competing interests.

Claire Jame performed the experiments, analyzed the data, approved the final draft.

Jérémy Tissier performed the experiments, analyzed the data, prepared figures and/or tables, authored or reviewed drafts of the paper, approved the final draft.

Olivier Maridet analyzed the data, approved the final draft.

Damien Becker conceived and designed the experiments, performed the experiments, analyzed the data, prepared figures and/or tables, authored or reviewed drafts of the paper, approved the final draft.

The following information was supplied regarding data availability:

The raw measurements are available in Tables 2–10 of the article.

List of specimens:

NMB = Naturhistorisches Museum Basel, Basel, Switzerland

NMBE = Naturhistorisches Museum der Burgergemeinde Bern, Bern, Switzerland

ML = Museum Langenthal, Langenthal, Switzerland

Pleuroceros pleuroceros (Duvernoy, 1853)

Referred material

Skull with right P1-M3 and left P2-M3 (original specimen lost, cast NMBE5031553, cast NMB-AS77)

Fragmented mandible with right p4-m3 and left m1–2 (original lost, cast NMBE5026739, cast NMB-AS78)

Right semilunate (original NMBE5031537, cast NMB-AS3)

Right McIV (original lost, cast NMB-AS79

Diaceratherium lemanense (Pomel, 1853)

Referred material

Skull with left M1–M3 (original exposed in ML, cast NMBE5031538, cast NMB-AS75)

Right maxillary fragment with P3-M3 (original NMBE5031539)

Right and left I1 (originals NMBE5031540)

Right hemi-mandible with i2 and p2-m3 (original NMBE5026738, cast NMB-UM6719)

Reconstructed incomplete mandible with left and right dental series with p2-m3 (original specimen exposed in ML, cast NMBE5031541, cast NMB-AS76)

Right femur (original NMBE5031542, cast NMB-UM6314)

Incomplete right tibia (original NMBE5031543)

Right tibia (original NMBE5031544, cast NMB-UM6315)

Right calcaneus (original NMBE5031545)

Two right astragali (original NMB-2017, original NMB-698)

Right MtII (original NMBE5026812)

Right MtIII (original NMBE5026812)

right I1: NMBE5031546

left i2: NMBE5031547

right P1: NMBE5031548

left P3: NMBE5031549

right P3: NMBE5031550

left lower cheek teeth: NMBE5031551

left lower cheek teeth: NMBE5031552.

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
