# Peer review of "Early Agenian rhinocerotids from Wischberg (Canton Bern, Switzerland) and clarification of the systematics of the genus Diaceratherium"

_PeerJ, doi:10.7717/peerj.7517_

## Round 0.1 · original submission · Minor Revisions

Dear authors,

We received three reviews of your manuscript, two suggested minor revisions and so prior to publication these minor revisions are necessary.
Please follow the comments of the reviewers and the suggestions they made in the annotated texts I attach to this mail.

Yours sincerely
Clara Stefen

·

Basic reporting

Latin names should be in italics; see for example line 212 (m. interossei), line 213 (m. extensor carpalis), line 433 (m. fibularis longus).

Experimental design

no comment

Validity of the findings

no comment

Additional comments

One of the best papers on rhinos I reviewed during the last year. Well-done.

·

Basic reporting

As a whole, the manuscript is well organized and clear, but there are some sentences or paragraphs for which I suggest some modifications in order to clarify the idea or the authors’ position (e.g., concerning systematics).
The Introduction is brief and synthetic, and I suggest not revealing the precise taxonomic results (see PDF).
Figures are relevant, of high quality and well described. I would only suggest alluding to them sometimes within description for some of the highlighted features.

Experimental design

The presented work is a review of fossil remains previously published many years ago, which leads to a new taxonomic interpretation of some of them. Some are figured for the first time. In addition, some comments on the controversial systematics of the genus Diaceratherium are also included. It is in this issue where I suggest a more detailed author’s opinion to avoid incongruences in the text (see PDF).
The review is based on morphological and metrical comparisons with the known European species of the epoch (Early Miocene).
I think metrical data could be gathered in one or more tables instead of being included within the descriptions, and maybe some of the compared specimens could be added to reinforce the affinities expressed in the text.
Concerning Table 3, it only shows the length of the metapodials of the discussed species. I think it would be more informative showing the whole available dimensions of each metapodial, both those studied (dimensions within the text) and those compared (see comment in PDF about Table 3).

Validity of the findings

The presented comparison (concerning morphology and size) supports the taxonomic results, but better explanations in the Discussion section are needed and tables could be improved to be more informative (see PDF). Particularly, the discussion on the validity of species of Diaceratherium needs to be expanded and reviewed to avoid incongruences.

Additional comments

I think all my suggestions do not imply major revisions of the manuscript, but most would be necessary before publishing it.
Only if the editor consider it necessary, I could evaluate the revised version.
The attached PDF shows all my comments and suggestions.

·

Basic reporting

The English expression is excellent, at least for the non-native English speaker I am. References are up-to-date and the state of the art correctly exposed. The structure is straightforward and rigorous. I have no comment about the figures and raw data are shared. Results are relevant, new, and well supported, based on the material described.

Experimental design

This manuscript meets the technical requirements of PeerJ and it matches its Aims and Scope. The research question is relevant and the results answer a 70 year-old question on the taxonomic identity of Wischberg rhinocerotids. Furthermore, this MS provides an interesting discussion on the genera Diaceratherium and Pleuroceros, with tantalizing perspectives at the European scale. I have found neither scientific nor technical flows and ethical issues are not into question. Methods are OK and the rationale is crystal clear.

Validity of the findings

I do agree with all the taxonomic assignments and commend the authors for solving this 70-year old puzzle. Figures and measurements are provided. Repeatability is ensured by clear and thorough description, following a proven descriptive protocol. Statistics are not applicable here (but see my suggestion on body weight estimates).

Additional comments

This is an interesting and useful work, providing new robust data on a classic locality from Switzerland. I have minor comments and suggestions aiming at improving the manuscript. You will find them below, sorted by order of appearance in the manuscript. I have also uploaded an annotated version of the merged pdf (mainly used for typos).

-line 36: please provide the original words abbreviated as USM;

-lines 100-108 (and Table 8 + Palaeoecology…): I would advise to be more careful with body weight estimates based only on lower molars, due to tremendous disparity in body shape and stoutness among rhinocerotids, likely to range from tapir-like to dachshund-like while having similar tooth size (Fortelius, M. 1990. In: J. Damuth and B.J. MacFadden (eds.), Cambridge University Press, Cambridge). It would be really useful to further provide estimates based on postcranial elements, such as long bones, metapodials (Fortelius, M. and Kappelman, J. 1993. Zoological Journal of Linnean Society, London 108: 85–101), and/or astragali (Tsubamoto, T. 2014. Acta Palaeontologica Polonica 59 (2): 259–265);

-lines 204-209: what is the general shape of the anterior side (high, low, narrow, wide)?

-lines 218-249 and 455-530: I would recommend to use your specimens from Wischberg as a reference and to overturn the corresponding sentences, such as “The referred specimens have a developed external groove and an angular trigonid on lower molars, contrary to representatives of Diaceratherium” (line 220).

-lines 360-361, 601 (and throughout the MS): please homogenize spelling (UK [centimetres / palaeobiogeographical] or US [characterized]);

-line 419 and throughout the MS (e.g., lines 517, 524):
“astragal” shall read "astragalus" or "astragali" (plural);

-line 447: does “lacking” mean broken or absent?

-line 517: shall read “collum tali”;

-Figure and table captions: to be complemented by providing a family-level assignment [“the rhinocerotid” or “(Perissodactyla, Rhinocerotidae)”] so that they are fully self-explanatory.

---

## Round 0.2 · accepted · Accept

Dear Authors,

You took all the comments of the reviewers into account and the manuscript is improved.